# Teach Less, Learn More: On the Undistillable Classes in Knowledge Distillation

**Yichen Zhu,**
Midea Group

**Ning Liu**
Midea Group

**Zhiyuan Xu**
Midea Group

**Xin Liu**
East China Normal University

**Weibing Meng**
Tsinghua University

**Yi Wang**
Midea Group

**Zhicai Ou**
Midea Group

**Jian Tang**[*]
Midea Group

## Abstract

Knowledge distillation (KD) can effectively compress neural networks by training a smaller network (student) to simulate the behavior of a larger one (teacher). A counter-intuitive observation is that a more expansive teacher does not make a better student, but the reasons for this phenomenon remain unclear. In this paper, we demonstrate that this is directly attributed to the presence of *undistillable classes*: when trained with distillation, the teacher's knowledge of some classes is incomprehensible to the student model. We observe that while KD improves the overall accuracy, it is at the cost of the model becoming inaccurate in these undistillable classes. After establishing their widespread existence in state-of-the-art distillation methods, we illustrate their correlation with the capacity gap between teacher and student models. Finally, we present a simple "Teach Less Learn More" (TLLM) framework to identify and discard the undistillable classes during training. We validate the effectiveness of our approach on multiple datasets with varying network architectures. In all settings, our proposed method is able to exceed the performance of competitive state-of-the-art techniques.

## 1 Introduction

More accurate machine learning models often demand more computation and memory at test time, making them difficult to deploy on computational-constraint devices. Knowledge distillation (KD) alleviates this burden by transferring the knowledge from an expensive teacher model to a cheap student model to improve the performance of this more efficient network. Over the past several years, we have witnessed a huge success in knowledge distillation and a surge of related literature on designing better distillation techniques.

An intuitive thought of knowledge distillation is that better teachers are supposed to make better students since a larger model can learn more visually meaningful and task-related information, and a student model can presumably benefit from learning with more informative knowledge. However, recent works show that such intuition is incorrect. In fact, prior works found that students distilled from a bigger teacher, one with more parameters and higher accuracy, can perform worse than the same students distilled from a smaller teacher [23, 9, 36, 25, 33]. The situation is getting more severe when training on a large-scale, challenging dataset such as ImageNet [5]. More surprisingly, the model with the same architecture [49] or even a model with lower performance [18, 46, 50] can be used as a teacher network to perform knowledge distillation, which plays a role of regularization [32, 24, 52]. Indeed, a large body of researches [55, 35, 53, 17, 34] devoted to solving this problem, but few minimal efforts have been put into understanding this phenomenon.

---

[*]Corresponding Author

36th Conference on Neural Information Processing Systems (NeurIPS 2022).

In this paper, we propose a new data-centric perspective on the phenomenon of "larger teacher, worse student". In contrast to the previous works, we cast the inefficacy of large teachers as a fundamental consequence of the class-dependent bias in KD. Specifically, we claim that:

*The undistillable classes are a direct result of the inefficacy of large teachers in distillation.*

Recall that prior works attribute the failure of large teachers in KD to the capacity mismatch between the teacher and student model. In other words, the student model cannot learn the teacher's knowledge due to their insufficient expressive power. Indeed, we find that their claim is true for *some* classes that posses bad *distillability*. We posit that these classes cause seemingly mismatched capacity over all samples in the dataset.

The empirical study supports our hypothesis. For samples from the undistillable classes, the feature representation similarity between the teacher and student model is extremely low compared to the regular classes. Furthermore, perhaps less surprisingly, the distilled model could obtain lower accuracy on these undistillable classes than their vanilla-trained counterpart. To further corroborate our theory, we show that the number of undistillable classes increases when the capacity gap between the two models increases.

After verifying our hypothesis, a natural question would be raised: to what extent do these undistillable classes exist? When discussing the property of knowledge distillation, previous literature typically evaluates vanilla knowledge distillation [14]. However, considering the evolution of distillation techniques over the past several years, we believe it is necessary to expand our study. To this end, we investigate the existence of the undistillable classes over three standard datasets (CIFAR100 [21], ImageNet1K [6], CUB-200 [42]) with more than 20 modern distillation techniques. We also include the evaluation of advanced architectures, i.e., vision transformer [39]. Our study shows that the undistillable classes generally exist, usually taking over from $10\%$ to $50\%$ of the total classes, depending on the distillation methods, datasets, and teacher-student pairs.

To resolve this issue, we present a novel distillation framework called "Teach Less, Learn More" distillation (TLLM). It is motivated by the successful teaching strategy in pedagogy[37], where teachers are encouraged to do less rote learning and provide more spaces for students to explore and discover their talents. Our methodology is conceptually similar: we aim to identify the undistillable classes and remove them entirely during distillation, allowing the student model to explore the representations of these classes with their own capacity. Specifically, we record the per-class teaching curve (distillation losses) and calculate its change over a moving window of K epochs. If the teaching curve for some particular classes is stagnated or upward, we consider the knowledge transfer process for these classes is no longer necessary. Then, we remove these classes in distillation (if it is an output-based KD, we replace them with the label smoothing technique to compensate for the regularization effect). Our approach, despite its simplicity, is proven to be effective in preventing the adverse effect brought by the undistillable classes and improving the overall accuracy.

Overall, we emphasize our contributions are the following: 1) we analytically show that the undistillable classes are the cause of inefficacy of the large teacher model; 2) we demonstrate that the accuracy of these undistillable classes drops after distillation, and our extensive experiments showing this is a general phenomenon, irrespective of distillation methods, datasets, or teacher-student pairs; 3) we propose a simple yet effective framework to identify and discard undistillable classes and illustrate the effectiveness our approach with extensive experiments.

## 2 On the Undistillable Classes in Knowledge Distillation

### 2.1 Formal Definition

In this section, we formally define the undistillable classes. Considering an multi-class predictor $F : \mathbb{R}^d \to \mathbb{R}^k$ as student and another predictor $G : \mathbb{R}^d \to \mathbb{R}^k$ as teacher. Giving a traing dataset $S = \{(x_i, y_i)_{i=1}^n\} \sim \mathbb{P}^n$ for distribution $\mathbb{P}$ over a set of instance $\mathcal{X}$ and label $\mathcal{Y} = [L] = [1, 2, \ldots, L]$. In the standard training, the goal is to approximate the risk $R(f; S)$ of functional class $f \in F$ via the empirical risk $\hat{R}(\mathbf{f}; S) = \frac{1}{N} \sum_n l(y, \mathbf{f}(x))$, for any bounded loss. For knowledge distillation, we can rewrite the distilled risk as $\tilde{R}(\mathbf{f}; S) = \frac{1}{N} \sum_n l(\mathbf{g}(x), \mathbf{f}(x))$, where the functional class $g \in G$.

The conventional setting consider all classes equally. Without loss of generality, we will consider the $S = S_r \bigcup S_u \sim \mathbb{P}^n$, where we split the data over distribution $\mathbb{P}$ into two categories, each category

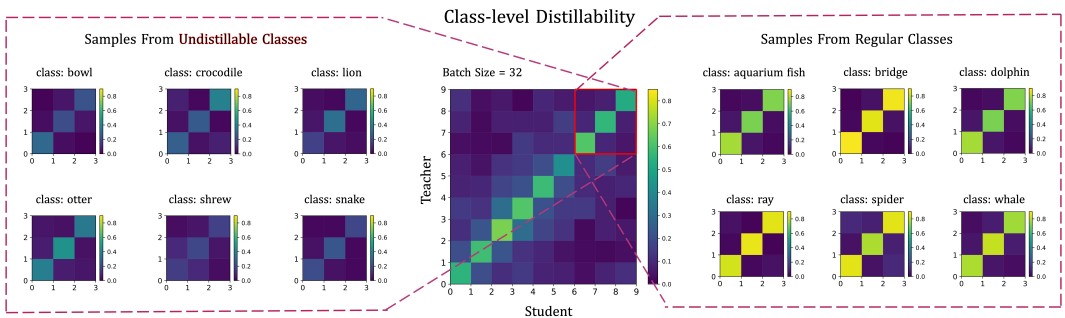

Figure 1: The representation agreement measured by CKA between teacher model and student model. For student network, the samples from undistillable classes clearly fail to match the representation from teacher model.

contains non-overlapping instances $S_u = \{(x_i, y_i^u)_{i=1}^n\}, \mathcal{Y}^u = [L] = \{L_{n+1}, L_{n+2}, \cdots, L\}, S_r = S \setminus S_u$ and $|\cdot|_c$ be a appropriate distillability measurement. In our context, we define the distillability function $|\cdot|_c$ as the difference in performance between the distilled student model with the train-from-scratch counterpart under certain evaluation metrics. Let $E(\cdot)$ be an evaluation metric that measures the performance of a particular multi-class predictor on $S$. Then, formally, we define $|\cdot|_c = E(F_{kd}) - E(F_{vanilla})$. Finally, we define that $S_u$ are a set of samples from the *undistillable classes* in distillation iff $|\mathbf{f}; S_u|_c \ll |\mathbf{f}; S_r|_c$.

## 2.2 Mismatched Representation for Undistillable Classes

The previous section formally defines the undistillable classes as classes with bad distillability. In this section, we develop a criterion to measure the distillability of a class quantitatively. Recall that the student is typically encouraged to mimic the behavior of the teacher model. Therefore, a converged student should impeccably match the teacher's behavior in an ideal setting. It implies that their feature representations should be the same as the teachers', given the same input on both models. As a result, we harness the similarity of feature representations between teacher and student to reflect the level of distillability of the student model. Inspired by Zhu et al. [55], we use the Center Kernel Alignment (CKA) [20] to calculate the representation similarity between two neural networks. To create a typical case of capacity mismatch, we follow the previous works [5] to use a ResNet56 as our pre-trained teacher model and a modified ResNet24 as the student model (see Appendix for details). We perform standard feature knowledge distillation [31] on stage four, which contains nine consecutive convolutional blocks. We choose to compare the network representation at convergence. For the CKA score figure, we primarily focus the score on the diagonal, which indicates the representation similarity of the convolutional layer at the same position in the network. In order to compare the representation similarity of samples from different types of classes, we deliberately choose 12 test samples from distinct classes. Note that we use samples from the test set to ensure that the observed behavior is not due to ad-hoc memorization.

In the middle of Figure 1, the feature similarity between teacher and student on each corresponding layer is presented. We can observe that the CKA scores for most layers are less than 0.6, and for some layers are even less than 0.5. These results indicate that despite the student indeed learning to match the teacher's representations, their features are far from perfect alignment. Notably, the representation similarity score is calculated based on the average score over a batch. Therefore, we decouple the matrices and check the CKA score for each sample individually. On the left side and right side of Figure 1, we give the CKA matrices at the last three layers for 12 samples. Each side contains six samples. We notice a clear trend: the CKA scores for samples from the left side are obviously much lower than those on the right side, indicating that the mismatched representations are not uniformly distributed over all samples. Therefore, we name the classes on the left side as *undistillable classes*. Our observation contradicts the conclusion from the prior work, which attributes the mismatched capacity as a general issue for all data [55, 5, 23]. We later show that at the left top of Figure 4, the accuracy for the undistillable classes drops after training with KD compared to their train-from-scratch counterpart.

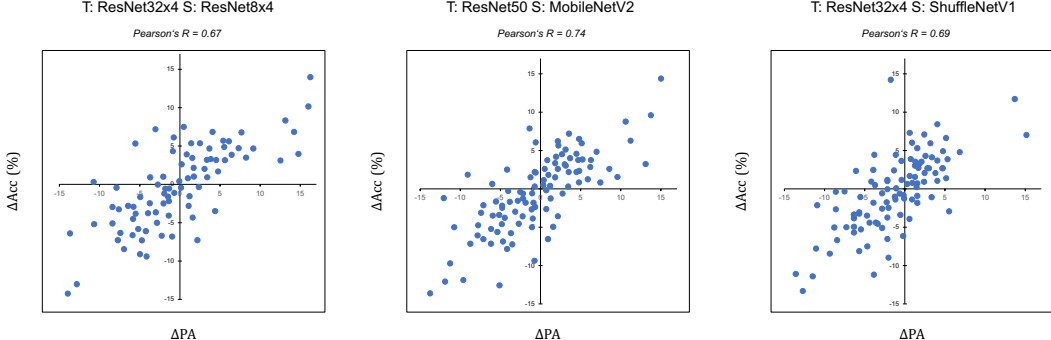

Figure 2: The $\Delta$PA versus $\Delta$Acc for three teacher-student pairs on CIFAR100. The Pearson's r is higher than 0.65 for all figures.

## 2.3 Bad Distillability hurts Accuracy

As aforementioned, a model would contain some classes with bad distillability. But what is the consequence of distilling knowledge to these classes? In this section, we show that the quality of KD (i.e., accuracy) is highly correlated to their distillability. Here we record the difference in accuracy between the vanilla trained model (training without distillation) and the model trained with knowledge distillation. Particularly, we adopt per-class accuracy:

$$\Delta\text{ACC}^m = \text{ACC}^m_{\text{vanilla}} - \text{ACC}^m_{\text{kd}} \tag{1}$$

where $m \in \mathbb{Y}$. The motivation behind this metric is straightforward: if the $\Delta\text{ACC}^m > 0$, then for $m^{th}$ class, the effect of KD is positive; otherwise, the KD hurts the student's performance on $m^{th}$ class. To demonstrate the correlation between per-class accuracy and per-class distillability, a naive approach is to compare the $\Delta$ACC with the CKA scores, as shown in the previous section. However, CKA measures per-layer feature representation similarity; simply averaging over the total number of layers may be inaccurate, not to mention that the teacher model usually contains more layers than students. Therefore, we present a new metric based on average prediction agreement:

$$\text{PA} := \frac{1}{n} \sum_{i=1}^{n} \mathbb{1} \, \text{argmax} \, \sigma(z_{t,i}) = \text{argmax} \, \sigma(z_{s,i}) \tag{2}$$

where $\sigma(z_t) = \frac{\exp(z_i)}{\sum_j \exp(z_i)}$ and $\sigma(z_s) = \frac{\exp(z_s)}{\sum_j \exp(z_s)}$ is the output of softmax function for teacher model and student model, respectively. Equation 2 is inspired by the fact that if the feature representations between two models are exactly the same, these two models ought to produce the same top-1 label. Similar to $\Delta$Acc, here we focus on the difference in prediction agreement $\Delta$PA between two models; one is trained from scratch, and another is trained with distillation:

$$\Delta\text{PA}^m := \text{PA}^m_{\text{vanilla}} - \text{PA}^m_{\text{kd}} \tag{3}$$

Specifically, we put these two metrics in a scatter plot chart for three varying teacher-student pairs on CIFAR100. Figure 2 reveals that for most of the classes with $\Delta$ACC $\leq 0$, its corresponding $\Delta$PA $\leq 0$ as well, indicating that if the student *become* disagrees with the teacher's prediction, the teacher's supervision can hurt student's performance on this particular class. In addition, we can observe that the level of disagreement is linearly correlated to the level of performance change: when measured with the Pearson's r, it obtained over 0.65 with a p-value less than 0.0005 on all three teacher-student combination, showing a statistically significant linear correlation between these two metrics. In conclusion, for any class in the student model, if it is undistillable, it certainly has $\Delta\text{ACC}^m < 0$.

## 2.4 The Larger the Capacity Gap, the More the Undistillable Classes

Now, we redirect our attention to the correlation between the level of capacity gap and the presence of the undistillable classes. We propose to leverage multiple teacher networks with varying capacity to distilled a single, low-capacity student. Specifically, we select the ResNet20 as our student model,

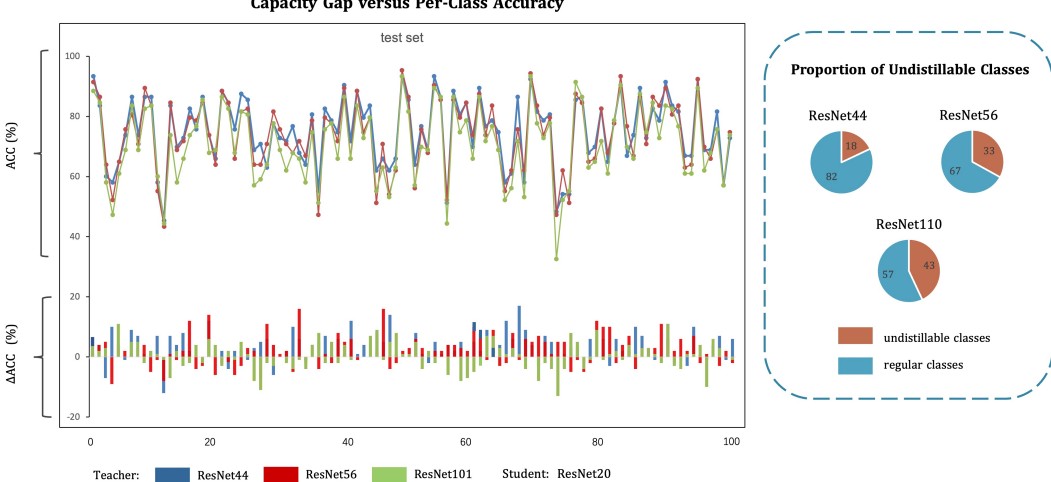

Figure 3: **Left:** Per-class ACC and per-class $\Delta$ACC for three teacher-student pairs in CIFAR100. **Right:** Proportion of Undistillable Classes distilling ResNet20 with three teacher models.

and pick ResNet44, ResNet56, and ResNet110 as three individual teacher model. For each set of experiments, we adopt one teacher model from the above to train a student model via vanilla KD [14]. This procedure mirrors that of mismatched capacity experiments in Cho et al. [5], except that our goal is to investigate the relationship of per-class accuracy with the capacity gap. As shown in Figure 3, we observe a clear relation between increase of the number undistillable classes, as increase in the capacity gap between models. Specifically, the number of total undistillable classes for the smallest student model (ResNet44) is almost tripled compare to the largest largest model (ResNet101).

## 2.5 On the Existence of the Undistillable Classes

The knowledge distillation is well-known for its "free" gain over compact models *on average* on an i.i.d test set. Modern distillation techniques only focus on improving the overall accuracy of the student model, with an intuition that the distillation is beneficial for all classes. Indeed, the overall accuracy is paramount, but the per-class accuracy is also important in many cases. Nevertheless, the distillation method inevitably brings class-dependent bias into the student model. As briefly shown in Figure 4, the distilled student model shows obvious class-dependent bias on all datasets. It is also worth noting that the accuracy drops dramatically for some undistillable classes. For instance, on ImageNet1K, there are 127 classes that lose over $4\%$ accuracy compared to their vanilla trained counterpart, and more than nine classes lose over $10\%$ on the test set.

In recent years, the techniques of distillation have evolved dramatically, i.e., feature-based distillation and self-supervision-based distillation [38]. Therefore, to ensure the undistillable classes are not a unique byproduct of the vanilla KD, we extend our experiments to include 26 state-of-the-art distillation frameworks, each of them evaluated by five different teacher-student pairs on CIFAR100. We categorize them into two groups (output-based KD and feature-based KD). Table 1 give a list of all methods that we used for evaluation. We only present the results for 13 of them due to limited space; the complete analysis and more evaluation of ImageNet-1K can be found in Appendix. Figure 5 shows the number of undistillable classes. It is surprising to show that all of them contain undistillable classes. For instance, the Resnet32x4:Resnet8x4 with vanilla KD has 52 undistillable classes, more the half of the total classes. Despite so many undistillable classes, all distillation methods improve the vanilla student model, some by a large margin. Such a phenomenon indicates that the effect of knowledge distillation is non-uniform at the class level. Moreover, the proportion of the undistillable classes is taking over from $10\%$ to $50\%$ over the total number of classes. Therefore, it is critical and necessary to control and reduce the number of undistillable classes in the student network.

Table 1: A summary of all distillation methods that we used for evaluation in Section 2.5

| Method | KD [14] | FitNet [31] | AT [47] | SP [41] | CC [29] | VID [1] | RKD [27] | PKT [28] | AB [13] | FT [19] | FSP [45] | NST [16] | CRD [38] |
|---|---|---|---|---|---|---|---|---|---|---|---|---|---|
| Pub | Arxiv'14 | ICLR'15 | Arxiv'16 | ICCV'19 | ICCV'19 | CVPR'19 | CVPR'19 | ECCV'18 | AAAI'19 | Neurips18 | CVPR'17 | Arxiv'18 | ICLR'19 |
| Output | ✓ | ✗ | ✗ | ✗ | ✓ | ✗ | ✗ | ✓ | ✗ | ✗ | ✗ | ✓ | ✗ |
| Feature | ✗ | ✓ | ✓ | ✓ | ✓ | ✓ | ✓ | ✓ | ✓ | ✓ | ✓ | ✗ | ✓ |

| Method | SSKD [43] | PAD [51] | WSKD [54] | LST [33] | OH [12] | TA [23] | ESKD [5] | SCKD [55] | KR [3] | SFTN [26] | CID [7] | FKD [11] | DKD [53] |
|---|---|---|---|---|---|---|---|---|---|---|---|---|---|
| Pub | ECCV'20 | ECCV'20 | ICLR'20 | ICLR'20 | ICCV'19 | AAAI'20 | ICCV'19 | ICCV'21 | ICCV'21 | Neurips'21 | Neurips'21 | ICLR'22 | CVPR'22 |
| Output | ✓ | ✓ | ✓ | ✓ | ✗ | ✓ | ✓ | ✓ | ✗ | ✓ | ✓ | ✓ | ✓ |
| Feature | ✗ | ✗ | ✗ | ✗ | ✓ | ✗ | ✗ | ✓ | ✗ | ✓ | ✗ | ✗ | ✗ |

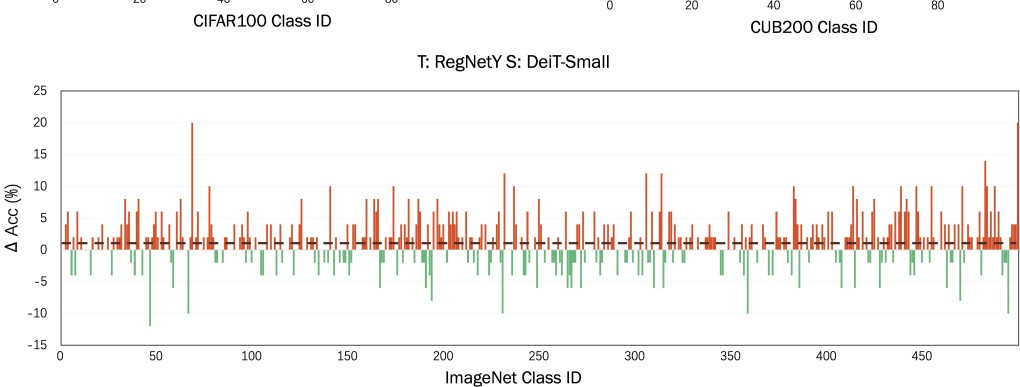

Figure 4: The per-class ΔACC of knowledge distillation on three datasets (CIFAR100 [21]/ImageNet1K [6]/CUB-200 [42]) with vanilla KD [14]. We only show the first 500 class in ImageNet1K and the first 100 classes in CUB-200. The dashed line denotes the improvement of average accuracy using KD.

# 3  Teach Less, Learn More (TLLM): Identifying and Discarding the Undistillable Classes

We have stressed the adverse effect and characterized the properties of the undistillable classes from the previous section. To this end, we give a simple framework, named "Teach Less, Learn More" distillation (TLLM), which analogy to a successful teaching principal in the pedagogy, to build better KD algorithms. Our idea is conceptually similar to the TLLM in the pedagogy, where the student is encouraged to have more autonomy and devise their own solutions to problems. In practice, the student model is supervised by both teacher's knowledge and the ground truth labels. Training with the model's task-oriented objectives is critical to the eventual performance [14], and provably to correct the wrong gradient directions that are brought by the teacher's mistaken knowledge [18]. Therefore, what we do is simple: identify the undistillable classes and then remove them from the distillation process. As a result, the student can learn these classes directly from the ground truth labels without interference.

The critical challenge of TLLM is how to find the undistillable classes during training. A naive yet computational expansive approach is first to train the model to convergence with distillation and then

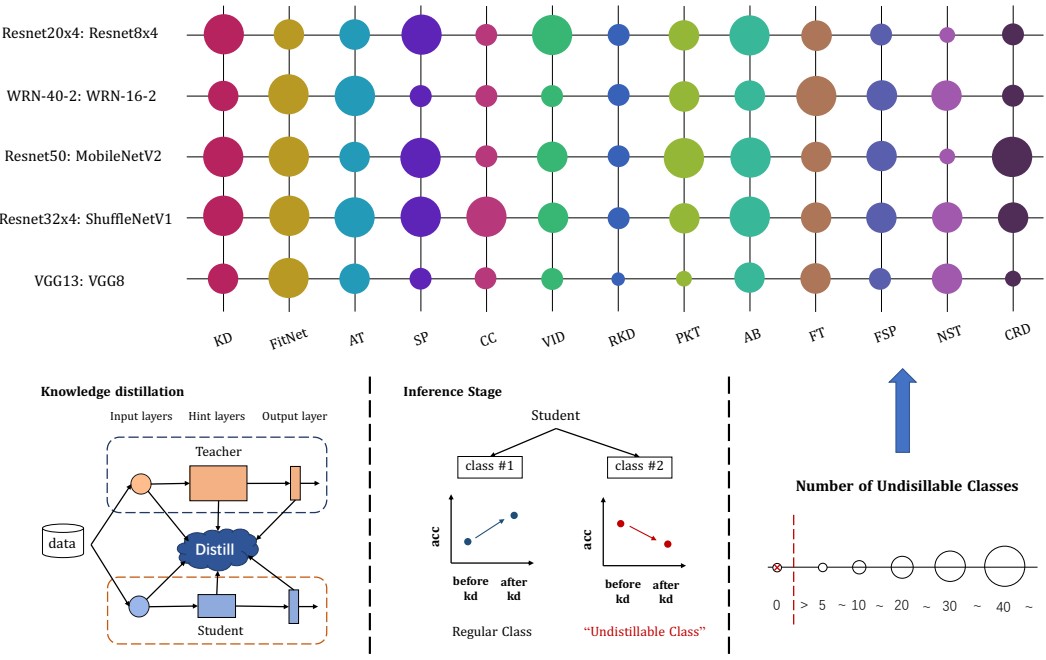

Figure 5: **Top**: the number of undistillable classes for diverse teacher-student pair evaluated by various distillation methods. More can be found in the Appendix. **Bottom left**: Three types of distillation approaches. **Bottom middle**: Definition of undistillable classes. **Bottom right**: Each size of circle represent a range of undistillable class number.

retrain it by manually eliminating the undistillable classes according to the performance on the test set. Instead, we propose to monitor the teaching curves (distillation losses) and set up a criterion to find the undistillable classes during training that are more computationally efficient.

Specifically, we pay attention to the per-class teaching curve, i.e., teaching curve $l(y_m, \mathbf{f}(x))$ on $m^{th}$ class, instead of the averaged distillation loss over the entire datasets. The teaching curve is calculated on the validation set. Then, we set a moving window over K epochs, which is started at $S^{th}$ epoch and end at $(S + K)^{th}$ epoch. During the distillation process, we record and update the per-class teaching curve $[l_S(y_m, \mathbf{f}(x)), l_{S+K}(y_m, \mathbf{f}(x))]$. If $l_S(y_m, \mathbf{f}(x)) - l_{S+K}(y_m, \mathbf{f}(x)) \leq \eta$, $0 \leq \eta$, meaning that the loss is moving upward in the K epochs moving window. Figure 3 gives an example of the teaching curve for an undistillable class. After training with approximately 50 epochs, the loss for this particular class suddenly jumps. Therefore, our method can capture such behavior, then identify and remove these undistillable classes during training. It is worth noting that these classes remain in the task-oriented objective for the student, making sure that the student can still learn from these data. Our approach is very efficient since the extra computational cost is linearly correlated with the number of classes, usually not a big number.

## 4 Experiment

### 4.1 Main Results

In this section, we evaluate our proposed method on CIFAR100 [21] and ImageNet [6] under various distillation settings. We refer reader to the Appendix for the implementation details.

**CIFAR100.** As aforementioned, our approach is orthogonal to other distillation techniques. Therefore, we investigate the performance of TLLM when combined with other methods. We report the experimental results on CIFAR100, which average over five trials. Specifically, we compare six distillation methods, and for each of them, we add our method to theirs. As shown in Table 2, our method consistently improve the overall accuracy for baseline. For instance, we improve the vanilla KD [14] by $2.54\%$. The performance gain mostly comes from the increased accuracy on undistillable

Table 2: Experimental results on CIFAR100 with CNN architectures. We observe that our approach improves overall accuracy and significantly reduces the number of undistillable classes.

| T-S Pair | ResNet32x4/ResNet8x4 (%) | | | ResNet50/MobileNetV2 (%) | | | ResNet32x4/ShuffleNetV1 (%) | | |
|---|---|---|---|---|---|---|---|---|---|
| Method | Vanilla | TLLM | Δ | Vanilla | TLLM | Δ | Vanilla | TLLM | Δ |
| Teacher | 79.42 | - | - | 79.34 | - | - | 79.42 | - | - |
| Student | 72.50 | - | - | 64.60 | - | - | 70.50 | - | - |
| KD [14] | 73.08 | **75.53** | +2.54 | 67.28 | **69.54** | +2.26 | 74.07 | **76.67** | +2.60 |
| FitNet [31] | 73.50 | **75.52** | +2.02 | 63.06 | **66.33** | +3.27 | 73.59 | **75.82** | +2.23 |
| AT [47] | 73.44 | **75.21** | +1.77 | 58.58 | **61.70** | +3.12 | 71.73 | **75.10** | +3.37 |
| OH [12] | 74.98 | **76.82** | +1.84 | 67.69 | **69.21** | +1.52 | 77.43 | **79.05** | +1.62 |
| KR [3] | 75.63 | **76.84** | +1.21 | 69.89 | **71.84** | +1.95 | 77.14 | **78.87** | +1.73 |
| SFTN [26] | 76.64 | **77.83** | +1.19 | 70.01 | **70.94** | +0.93 | 79.58 | **80.04** | +0.46 |

classes. Later, we will show that our approach also reduces the number of undistillable classes. It is also surprising to find that some methods that were once inferior to other approaches, i.e., the accuracy of OH [12] is $0.65\%$ lower than KR [3] on ResNet32x4/ResNet8x4, obtain higher accuracy after applying TLLM, showing that many "outdated" algorithms have the potential to be quality methods. Moreover, our approach is practical for both output-based (i.e., KD [14])and feature-based distillation [31, 3, 47] algorithms.

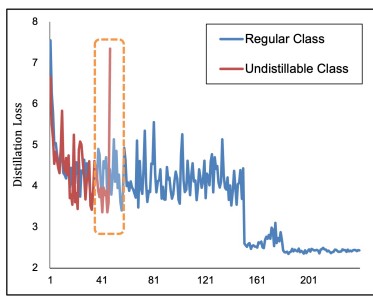

Figure 6: Teaching curve for regular class and undistillable class.

**ImageNet.** ImageNet is a large-scale classification dataset that contains 1.28 million images for training and 50k images for validation. Unlike conventional papers that only report the performance of CNN models, we also consider vision transformer (ViT), an advanced network architecture that has drawn increasing attention lately.

For CNN-based architectures, we evaluate two teacher-student pairs: 1) ResNet32 as a pretrained teacher and ResNet18 as a student model, a widely adopted setting follows CRD [38], 2) ResNet50 as a pretrained teacher, and MobileNetV1 as a student model, a less conventional setting that some papers have used. The experimental results are reported in Table 7. Our TLLM, which is built based on KR [10], achieves a considerable improvement over its baseline. It is also worth mentioning that the performance of TLLM outperforms other state-of-the-art distillation methods.

For ViT architectures, we evaluate two types of teacher-student pairs: 1) the RegNetY [30] and ViT-Tiny/ViT-Small [8] pairs are the popular setting in DeiT [39] which apply a CNN model as a teacher network to distill a vision transformer, 2) the CaiT-S23 [40] and ViT-Tiny pairs adopt an advanced ViT model as a teacher, whose performance is slightly inferior to RegNetY. We compare with DeiT [39], a benchmark that uses teachers' hard labels as dark knowledge to supervise the student network. We notice that in Table 4, we achieve higher accuracy than DeiT on all teacher-student pairs. Particularly, when the capacity gap between teacher and student is large, i.e., RegNetY/ViT-Tiny and CaiT-S24/ViT-Small, the improvement in the validation set is more notable.

## 4.2 Ablation Study

**Number of the Undistillable classes.** Our method specifically identified the undistillable classes and then discarded them during distillation. Therefore, we conduct experiments to check whether our approach reduces the number of undistillable classes. In the Appendix, we show that our proposed TLLM significantly reduces the number of undistillable classes in the student models, allowing balanced performance improvement.

Table 3: Experimental results on ImageNet1K with CNN architectures.

| Teacher | Baseline | Distillation Framework (%) | | | | | | | | | |
|---|---|---|---|---|---|---|---|---|---|---|---|
| ResNet32 | ResNet18 | KD [14] | AT [47] | OFD [12] | CRD [38] | SRRL [44] | KR [3] | SFTN [26] | CID [12] | DKD [53] | **TLLM** |
| 73.3 | 69.8 | 70.7 | 70.7 | 71.2 | 71.2 | 71.7 | 71.6 | 71.1 | 71.9 | 71.7 | **72.6** |
| ResNet50 | MobileNetV1 | KD | AT | OFD | CRD | SRRL | KR | SFTN | CID | DKD | **TLLM** |
| 76.2 | 68.9 | 70.5 | 69.6 | 71.3 | 71.4 | 71.7 | 72.6 | 71.5 | 72.3 | 72.1 | **73.2** |

Table 4: Experimental results on ImageNet1K with vision transformer architectures.

| T-S Pair | RegNetY/ViT-Tiny (%) | | RegNetY/ViT-Small (%) | | CaiT-S24/ViT-Tiny (%) | | CaiT-S24/ViT-Small (%) | |
|---|---|---|---|---|---|---|---|---|
| Method | Vanilla | TLLM | Vanilla | TLLM | Vanilla | TLLM | Vanilla | TLLM |
| Teacher | 82.9 | - | 82.7 | - | 82.7 | - | 82.7 | - |
| Student | 72.2 | - | 79.9 | - | 72.2 | - | 79.9 | - |
| KD/DeiT [39] | 74.5 | $\mathbf{76.3}_{+1.8}$ | 81.2 | $\mathbf{81.9}_{+0.7}$ | 74.4 | $\mathbf{76.1}_{+1.7}$ | 81.3 | $\mathbf{82.0}_{+0.7}$ |

**Does TLLM allow larger teacher to make better student?** Our approach is motivated to resolve the "larger teacher, worse student problem"; thus, it is natural to verify whether our approach can solve this issue. In the Appendix, we show that our approach allows the larger teacher to distill students with higher accuracy.

## 5  Analysis of the Undistillable Classes

This section will thoroughly analyze the undistillable classes we observed in the previous empirical study.

### 5.1  Hard Classes of *Student* $\neq$ Undistillable Classes

One may wonder whether the undistillable classes can be represented by the hard classes in the student model (classes with relatively low accuracy). This is a reasonable hypothesis since challenging classes of student networks (trained without distillation) can be hard to gain more knowledge from the teacher. Nevertheless, we show that undistillable classes are not the hard class in the student model. Specifically, we consider three varying teacher-student pairs: ResNet32x4/ResNet18, WRN-40-2/WRN-16-1, and ResNet56/ResNet20 on CIFAR100 dataset that is trained with vanilla KD [14]. We compare the test accuracy of the student model with the improvement in test accuracy with the student model train with and without KD. Figure 7 present the results.

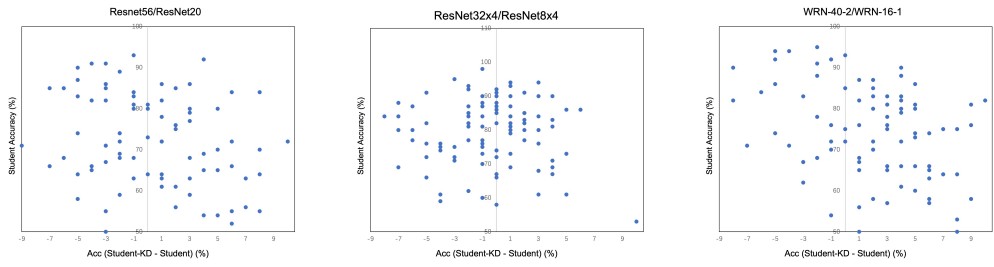

Figure 7: Hard classes of student model are not undistillable classes

We can observe that the correlation between students' accuracy and the improvement of KD is almost random, showing that the hard classes of a student are not equivalent to undistillable classes.

### 5.2  Hard Classes of *Teacher* $\neq$ Undistillable Classes

Another reasonable hypothesis is that the challenging classes in teacher model can be undistillable classes because teacher model is normally imperfect. For instance, ResNet-50 can only achieve 76.2% on ImageNet dataset. Can we find undistillable classes by looking at the hard classes of the teacher model? To answer this question, we conduct similar experiments as in the previous section. The only difference is that we record the teacher's test accuracy instead of recording the student's test

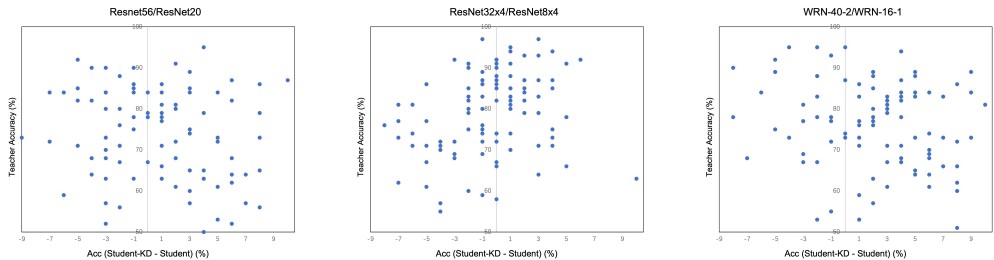

Figure 8: Hard classes of teacher model are not undistillable classes

accuracy. Also, we conduct experiments on three teacher-student pairs. Figure 8 presents the results, which lead to the same conclusion that hard classes of teacher model are not undistillable classes.

# 6   Related Work

The most correlated works are the research discussing the relationship between teacher and student. Many works observe that the performance of teacher networks does not necessarily improve the performance of its distilled student. Cho et al. [5] is the first work to present this particular phenomenon and argue that the capacity mismatch between teacher and student gives rise to this behavior. They propose applying the early stopping technique to train the teacher, and such an early-stopped teacher can surprisingly improve students' accuracy. It was later proved that early-stopped teachers improve data efficiency during distillation [18]. Mirzadeh et al. [23], inspired by BAN [9], show that using single or multiple intermediate networks, which are larger than the student yet smaller than the teacher, can also narrow the capacity gap. Zhu et al. [55] propose an adaptive distillation method based on the observation that the status of the capacity gap change constantly during training. Lukas et al. [22] find out that teachers trained with label smooth severely affect the student's accuracy if vanilla KD is used. Although this opinion is disputed by Shen et al. [33], following work demonstrate that the label-smoothed model can be an effective teacher if and only if some strict rules are applied, i.e., assign a large number to temperature term [2]. Other works see that even a bad teacher can teach a good student because self-distillation is provably a regularization technique [52, 24, 49, 15]. Nevertheless, none of these works investigate the capacity mismatch problem from the class level.

# 7   Conclusion

This paper studies the "larger teacher, worse student" phenomena in knowledge distillation. Unlike prior works, which attribute the inefficacy of large teachers as a capacity mismatch issue on samples, we consider the presence of undistillable classes as the direct cause of mismatched capacity. Our paper verified that these classes lower the ability of the student model to match the teacher's feature representation and predictive distribution, making the accuracy drops considerably in some arbitrary classes, therefore, hurting the overall performance. We further demonstrate that the existence of undistillable classes is universal, irrespective of distillation methods, datasets, and student-teacher configurations. Finally, we propose a "teacher less, learn more" (TLLM) method to alleviate this issue by targeting and removing samples from the undistillable classes. Our experiments have shown that our TLLM can greatly eliminate the adverse effect of undistillable classes to improve the overall performance significantly. Overall, we believe that our analysis and proposed method provides an interesting and practical new perspective on designing distillation algorithms.

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
