## A    Limitations and Potential Negative Social Impacts.

**Limitation.** Our work investigates the "larger teacher, worse student" phenomena in knowledge distillation through the lens of the undistillable classes. However, we only discuss image classification. Therefore, we do not guarantee the validity of our observation on other tasks, i.e., object detection, and cannot guarantee our approach's effectiveness on these tasks. We propose a method to resolve this issue in Section 4. We have extensively verified our approach's effectiveness in the experimental section but may still fail in other untested scenarios. Also, both our phenomenon and solutions lack theoretical justification and may leave to the future work to solve.

**Potential Negative Social Impacts** This paper presents the concept of undistillable classes, which can cause potential negative social impacts. Specifically, we found that knowledge distillation can hurt the performance of arbitrary classes. In addition, these classes can be sensitive, i.e., gender or race, which can negatively impact society. In Section 3, we propose a solution to mitigate this problem but cannot fully address them. We hope future work can completely resolve this issue.

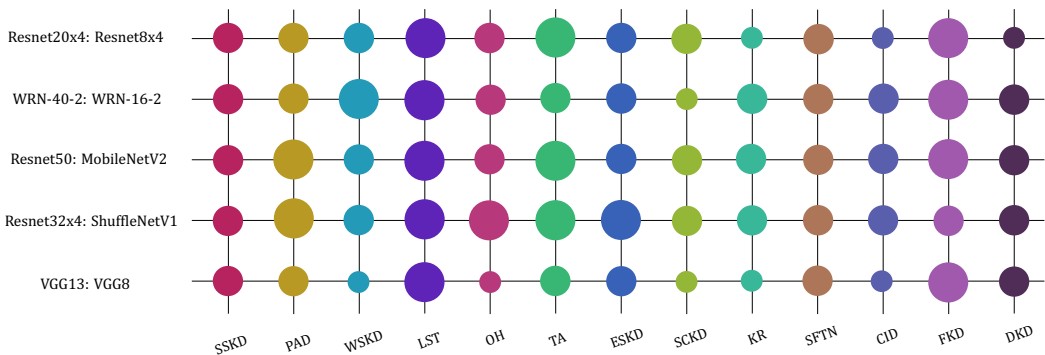

Figure 9: The number of undistillable classes for five teacher-student pair evaluated by additional 13 distillation methods.

## B    Implementation Details

**Implementation Details.** For all experiments in CIFAR100 [21], we follows the implementation in CRD [38] (https://github.com/HobbitLong/RepDistiller). For those methods that are not implemented in CRD, we follow the implementation in the open-sourced codes and transplant their approach to CRD. Since most of these method provides hyper-parameters for CIFAR100, we do not modify them. For experiments on CUB200 [42], because few KD methods open-sourced their implementations on this dataset, we leverage the same hyper-parameters as in CIFAR100. For all $\eta$ in the experiments, we conduct a hyper-parameter search on the validation dataset. During training, we randomly select 5% of the original training data as validation data to find the undistillable classes. We ensure the same number of samples are selected for each class. The best $\eta$ is obtained through hyper-parameter search, where we split another 5% of the pruned training data as validation data. The test data is not involved during neither training or hyper-parameter search stage. The experiments on ImageNet is similar, except that we only select 2% of them since ImageNet contains more training data than CIFAR100.

For all experiments in ImageNet1K [6], we use different implementations. For vision transformer architectures, we follows the DeiT [39] and CaiT [40] (https://github.com/facebookresearch/deit) to conduct our analysis. For other experiments with CNN architectures, we follows the implementation of mdistiller (https://github.com/megvii-research/mdistiller), we transplant some of the open-sourced methods, i.e., SRRL [44], SFTN [26], CID [7], to this platform for fair comparison.

**Details in Section 2.2.** In Section 2.2 we use modified ResNet24 as student to perform KD on a ResNet56 teacher model. Specifically, we ensure that the last stage in ResNet24 has the same number of layers as ResNet56 for fair comparison, since the resolution of the image are the same.

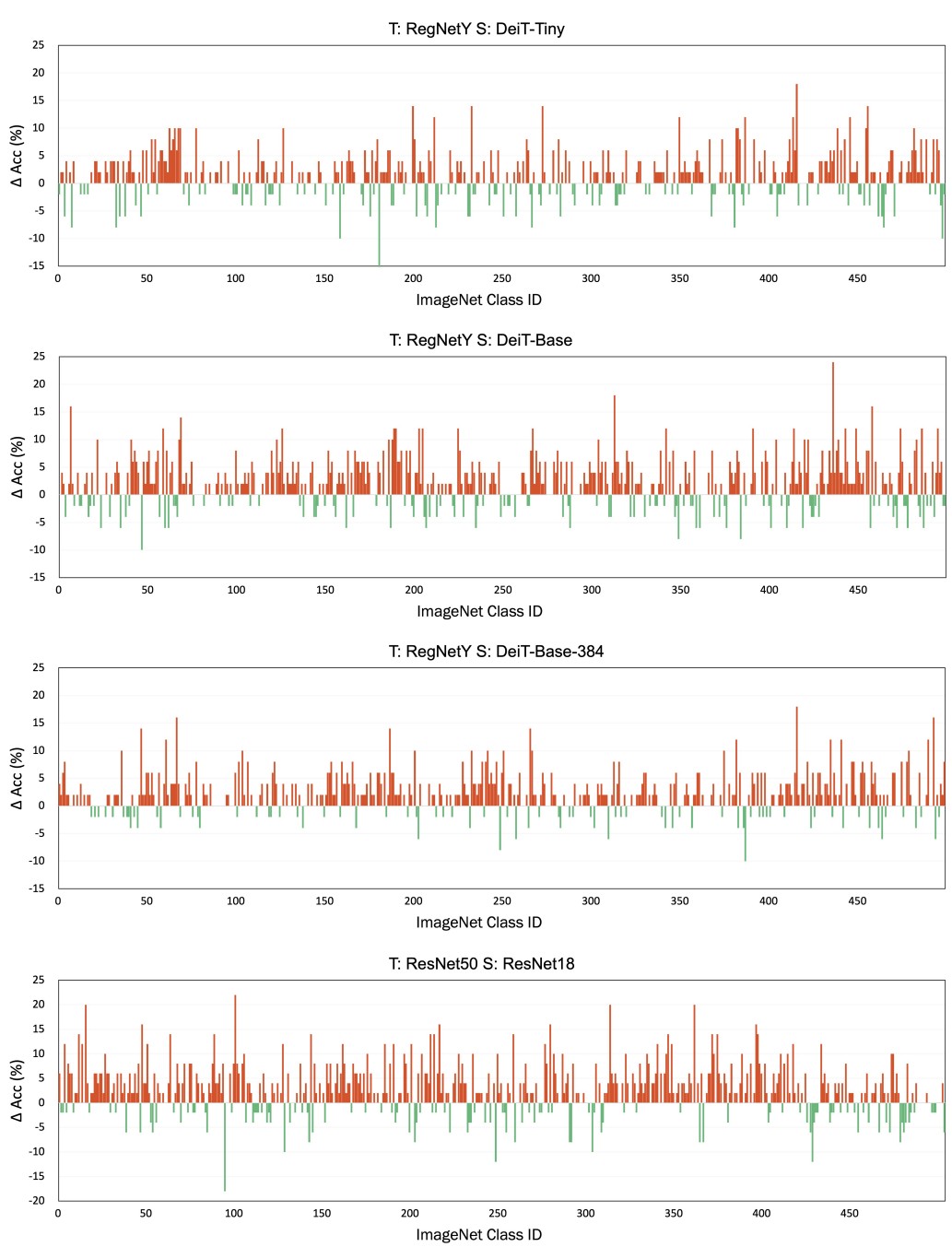

Figure 10: The per-class $\Delta$ACC of knowledge distillation on ImageNet1K.

## C The Undistillable Classes: More Evidences

We have mentioned the existence of the undistillable classes in general to various methods, and Table 1 gives a comprehensive list of methods for which we studied. To complete the analysis, Figure 9 presents the number of undistillable classes on CIFAR100 with five teacher-student pairs on another thirteen methods. We notice that for most approaches, there are more than 30 classes that are undistillable. We observe no clear advantage in feature-based distillation (OH and KR) compared to output-based distillation. Prior works that have proven capably bridge the capacity gap between teacher and student also demonstrate no superiority on this metric. We think existing methods need

Table 5: Experimental results on CIFAR100 which demonstrate that TLLM reduced number of undistillable classes of the baseline

| T-S Pair | ResNet32x4/ResNet8x4 (%) | | | ResNet50/MobileNetV2 (%) | | | ResNet32x4/ShuffleNetV1 (%) | | |
|---|---|---|---|---|---|---|---|---|---|
| Method | Vanilla | TLLM | Δ | Vanilla | TLLM | Δ | Vanilla | TLLM | Δ |
| KD [14] | 52 | 16 | -36 | 48 | 14 | -34 | 54 | 15 | -39 |
| FitNet [31] | 39 | 14 | -25 | 46 | 17 | -29 | 45 | 19 | -26 |
| AT [47] | 37 | 17 | -20 | 39 | 13 | -26 | 43 | 17 | -26 |

Table 6: Larger teacher makes better student with TLLM.

| Student | Teacher | | | |
|---|---|---|---|---|
| WRN16-2 | W-28-2 | W-40-2 | W-16-4 | W-28-4 |
| 73.43 | 75.40 | 75.72 | 77.31 | 78.56 |
| KD | 75.21 | 74.97 | 75.38 | 75.21 |
| TLLM | 76.02 | 76.54 | 77.11 | 77.35 |

to take account of this phenomenon more explicitly and carefully. Another observation is that the capacity gap indeed plays a vital role in the KD. We notice that the capacity gap between ResNet50: MobileNetV2 pair and the ResNet32x4: ShuffleNetV1 pair is larger than in other pairs. Thus for some methods, there are more undistillable classes in these pairs.

We further showcase of vision transformer on ImageNet1K. Besides the DeiT-small, as we showed in Figure 4 (bottom), we further provide the ΔACC on the first 500 classes on ImageNet1K for DeiT-tiny, DeiT-base, and DeiT-base with 384 resolution on testing. For CNNs, we provide ResNet50-ResNet18 as teacher-student pairs. As shown in Figure 10, we consistently observe the class-dependent phenomenon.

# D   Ablation Study

**TLLM reduce the Number of the Undistillable classes.** Table 5 shows that our TLLM indeed reduce the undistillable classes.

**Does TLLM allow larger teacher to make better student** Table 6 presents experiments to verify that our proposed TLLM can indeed improve the student's accuracy when using a large teacher. We see that the vanilla KD achieves lower accuracy for WRN-16-2 on a larger teacher than a smaller teacher, yet our approach can let large teachers make better students.

# E   More Analysis of the Undistillable Classes

This section will thoroughly analyze the undistillable classes we observed in the previous empirical study.

## E.1   Are Undistillable Classes Network-Independent?

In this section, we study whether undistillable classes are network-independent. We define the network-independent here as different teacher-student pairs. We consider three teacher-student pairs in our experiments, WRN-40-2/WRN-16-2, ResNet56/ResNet20, and ResNet32x4/ResNet8x4. We find out the undistillable classes are network-dependent, as shown in Figure 11

## E.2   Are Undistillable Classes Capacity-Gap-Independent?

Given the same teacher model, will the student model have different undistillable classes? In Figure 7 we present five classes that are undistillable classes in four different student architectures, where each of these models is distilled by a RegNet-Y model. Note that these are the only five classes

Table 7: Five classes in DeiT based student that consistently hurt by KD.

| ID | Name | ΔACC (%) | | | | |
|---|---|---|---|---|---|---|
| | | DeiT-Tiny | DeiT-Small | DeiT-Base | DeiT-Base-384 | Avgerage |
| 212 | English Setter | -4 | -2 | -8 | -2 | -4 |
| 356 | Weasel | -2 | -4 | -2 | -4 | -3 |
| 638 | Maillot | -8 | -24 | -10 | -14 | -14 |
| 744 | Missile | -12 | -18 | -18 | -8 | -14 |
| 876 | Tub | -4 | -8 | -6 | -8 | -6.5 |

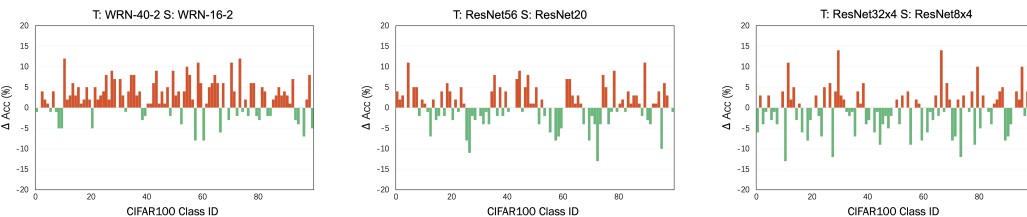

Figure 11: Hard classes of teacher model are not undistillable classes

that are undistillable across four teacher-student pairs. It is perhaps not surprising that most of the undistillable classes are dependent on the capacity gap between the teacher model and the student model

### E.3 Are Undistillable Classes Distillation-Independent?

We further explore whether undistillable classes depend on the distillation method. In Figure 12, we compare four different distillation method on the same teacher-student pair (ResNet32x4/ResNet8x4) on CIFAR100. We observe no clear trend that some classes are consistently undistillable across different distillation methods.

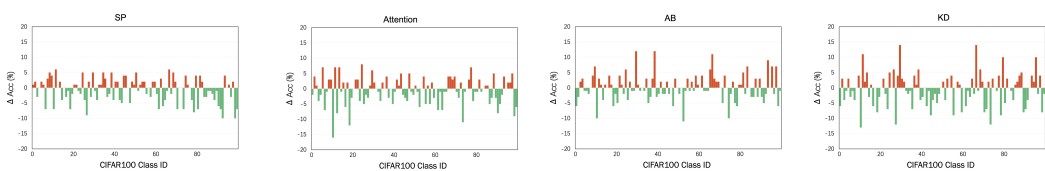

Figure 12: Undistillable classes are distillation dependent

### E.4 Visualization of Undistillable Classes versus Distillable Classes

To give a more intuitive understanding of undistillable classes, we utilize t-SNE to visualize the penultimate layer representations of ResNet8x4 on CIFAR-100 trained with ground truth labels and distillation. Figure 13 presents the visualization. We can observe that those clusters with clean boundaries (meaning that their prediction is accurate) can be either distillable or undistillable classes. It again verifies our previous conclusion that we can not tell whether a class is distillable or not based on their relative accuracy on the student model.

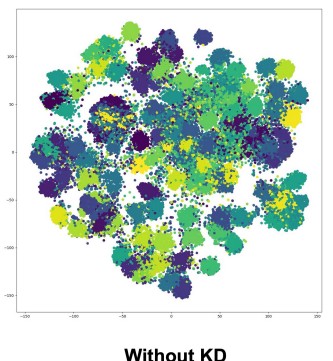 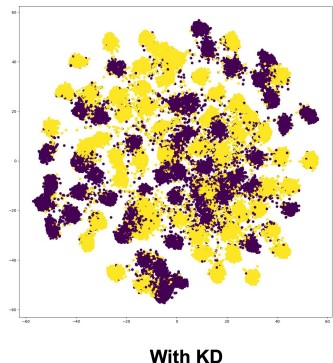

**Without KD**        **With KD**

Figure 13: Visualizing of the penultimate layer representation of ResNet8x4 on CIFAR-100 training set using t-SNE. Left: training with ground truth label. Right: training with knowledge distillation. The purple color represents undistillable classes, and the yellow color denotes the rest of the classes.

# F  More Experiments of TLLM

## F.1  More Experiments on ImageNet

In Table 2, we test our approach on CIFAR100 based on multiple baseline. In this section, we conduct a similar experiments to show that our proposed TLLM can consistently improve the baseline method on large-scale dataset. We conduct three experiments based on KD, AT, and SFTN on ImageNet with two different teacher-student pairs. Our results are presented in Table 8, which indicates the TLLM is effective on these KD methods.

Table 8: Experimental results on ImageNet with CNN architectures. The TLLM is performed on multiple baseline. Our approach consistently improve the state-of-the-art KD approaches.

| T-S Pair | ResNet32/ResNet18 (%) | | | ResNet50/MobileNetV1 (%) | | |
|---|---|---|---|---|---|---|
| Method | Vanilla | TLLM | $\Delta$ | Vanilla | TLLM | $\Delta$ |
| Teacher | 73.3 | - | - | 76.2 | - | - |
| Student | 69.8 | - | - | 68.9 | - | - |
| KD [14] | 70.7 | **72.1** | +1.4 | 70.5 | **72.0** | +1.5 |
| AT [47] | 70.7 | **71.5** | +0.8 | 69.6 | **70.6** | +1.0 |
| SFTN [26] | 71.1 | **72.0** | +0.9 | 71.5 | **72.8** | +1.3 |

## F.2  Comparison with Label Smoothing Baseline

In TLLM, the label smoothing techniques are used when the undistillable classes are discarded during distillation process, in order to compensate for the regularization effect that was brought by the KD. Therefore, we compare our approach to the baseline that is trained with label smoothing for fair comparison. We adopt the same hyper-parameter for all experiments (including TLLM), which set $\alpha = 0.1$. Table 9 present the results of comparing TLLM with label smoothing baseline. For fair comparison, our TLLM is ran based on KD [14]. We observe that the label smoothing indeed improve the baseline method. Nevertheless, it is much less effective than KD, which is consistent with the observation in OLS [48]. For instance, on ResNet50/ShuffleNetV1, the label smoothing only improve the student model by 0.28%, while our method increase the top-1 accuracy by 6.17%. In conclusion, we believe this experiment is sufficient to show that the improvement of TLLM cannot solely attribute to the label smoothing techniques.

Table 9: Comparison with label smoothing baseline on CIFAR-100. The LS denotes the label smoothing technique. We also compare with KD [14]. Our TLLM is developed based on KD [14]. All LS experiments use hyper-parameter with 0.1, including TLLM.

| T-S Pair | ResNet32x4/ResNet8x4 | | | | ResNet50/MobileNetV2 | | | | ResNet50/ShuffleNetV1 | | | |
|---|---|---|---|---|---|---|---|---|---|---|---|---|
| Method | Student | KD | LS | TLLM | Student | KD | LS | TLLM | Student | KD | LS | TLLM |
| Acc (%) | 72.50 | 73.08 | 73.86 | 75.53 | 64.60 | 67.28 | 65.11 | 69.54 | 70.50 | 74.07 | 70.78 | 76.67 |

## F.3 TLLM on CUB-200-2011

**Experimental setting.** We set batch size to 64 with initial learning rate of 0.05, the learning rate scheduler is set with decay factor of 0.1 after every 30 epochs to train each model for 120 epochs. We use random cropping, brightness jitter and random flip data augmentations, a standard training setting by MXNet [4]. For KD [14] baseline, we set $\alpha = 0.5$ and temperature $T = 3$. Table 10 gives the experimental results. We can observe a clear advantage of our method over vanilla KD [14]. For example, we improve the KD by 1.19% on ResNet50/ResNet18 teacher-student pairs.

Table 10: Experimental results on CUB-200-2011.

| Dataset | CUB-200-2011 | | | | | |
|---|---|---|---|---|---|---|
| T-S Pair | ResNet50/ResNet18 (%) | | | ResNet50/MobileNetV2 (%) | | |
| Method | Vanilla | TLLM | Δ | Vanilla | TLLM | Δ |
| Teacher | 82.17 | - | - | 82.17 | - | - |
| Student | 76.89 | - | - | 79.20 | - | - |
| KD [14] | 79.96 | **81.15** | +1.19 | 81.18 | **82.26** | +1.08 |

## F.4 Comparison with ESKD

We stress that there are two early-stopping based KD algorithms in ESKD. The first one is the adopt early-stopping *during distillation*, and the second one is the use early-stopping to *train teacher model*. Despite that the latter was emphasis by the ESKD and it is also the commonly referred method, the former was more correlated to our method. Therefore, we compare our approach to the prior training strategy instead of the latter for fair comparison. Notice that ESKD require hard threshold to stop the distillation process during student's training stage. We conduct a hyper-parameter search which stopped the distillation process at 70%, 80%, 90%, and 95% of the total training epochs and report the best results. Table 11 present the experiments on CIFAR100 and ImageNet1K datasets. On CIFAR100, we use ResNet50 as teacher model and MobileNetV2 as student model. On ImageNet, we use ResNet34 as teacher model and ResNet18 as student model. It is clear that ESKD can improve the baseline method, nevertheless, it is much less effective than our proposed TLLM. We think the reason behind such huge performance gap between TLLM and ESKD is because the class-wise KD perspective.

Table 11: Direct Comparison with ESKD [5].

| Dataset | CIFAR100 | | | ImageNet1K | | |
|---|---|---|---|---|---|---|
| Teacher-Student Pair | ResNet50/MobileNetV2 (%) | | | ResNet34/ResNet18 (%) | | |
| Method | Vanilla | ESKD | TLLM | Vanilla | ESKD | TLLM |
| Top-1 Accuracy (%) | 64.6 | 66.4 | **69.5** | 69.8 | 70.7 | **72.1** |