# OpenReview forum: "Teach Less, Learn More: On the Undistillable Classes in Knowledge Distillation"
_NeurIPS.cc/2022/Conference — NeurIPS 2022 Accept_

### Official Review · Reviewer_NdAQ · 2022-07-06

**Rating:** 5
**Confidence:** 4
**Soundness:** 2 fair
**Presentation:** 3 good
**Contribution:** 2 fair

**Summary:**

The major contributions of this paper are 2-fold:
1) The thesis of this paper is to study the “larger teacher, worse student” phenomenon in knowledge distillation (KD) from a data-centric approach. The authors show that the above phenomenon happens due to the presence of undistillable classes in the teacher. This observation is shown across different KD methods, datasets and teacher-student architectures.

2) The authors propose a simple, yet effective fix -- Teach Less, Learn More (TLLM) -- to identify and discard undistillable classes during student training, thereby improving the final accuracy of students.


**Questions:**

Please see Weaknesses section above for a list of questions. Further please consider answering the following questions:

1) What is the final optimization / loss function (preferably in equation form)?
2) What is the mixture parameter ($\alpha$) used for label smoothing in undistillable classes?


**Limitations:**

Limitations / potential societal impacts discussed in Supplementary Section A.

**Strengths And Weaknesses:**

**Strengths:**
1) This paper is written / presented clearly (especially Fig 5, bottom Middle). It is easy to follow.
2) The proposed TLLM framework shows noticeable improvements on CIFAR-100 (across different KD methods, teacher-student architectures).


**Weaknesses:**
1) Although the work is technically sound, do you think the explanation / definition of undistillable classes is a bit overkilled? Can the following be a more simpler explanation (I will use ImageNet experiment as example): The teacher network is not perfect. That is the top1 accuracy of ResNet-50 teacher is 76.2% (Table 3). Therefore, the teacher tends to make a noticeable amount of errors. If class-wise analysis is performed on the teacher based on misclassified samples, it corresponds to a subset of classes which can be treated as undistillable classes. Therefore, the proposed TLLM avoids such situations by eliminating classes where the teacher's predictions are largely incorrect and replacing them with ground-truth labels (one-hot vectors). Some analysis I’d like to see are:
- What is the average accuracy of ResNet-50 teacher (Table 3) for regular and undistillable classes?
- Can you decompose the improvements obtained ($\Delta$) in Tables 2, 3 for regular and undistillable classes?

2) ImageNet results are insufficient to assess the efficacy of proposed TLLM: What is the reason for using only KR (Chen et. al) as the baseline for ImageNet experiments? Can the authors produce a table similar to Table 2 for ImageNet to show the improvements of the proposed TLLM scheme?

3) The Baseline performance of MobileNetV2 in Table 3 (ImageNet) seems to be very low. top1 accuracy should be ~72.154% whereas the reported accuracy is only 68.900%. Please clarify. Link to public pytorch models with top1 accuracies: https://pytorch.org/vision/stable/models.html

4) If the authors use label smoothing for undistillable classes (Lines 66-68) in the student, it is not reasonable to compare with reported Baselines as label smoothing will further improve accuracy (by alleviating models’ overconfidence [1, 2, 3]). A more reasonable comparison will be to include the accuracies of students trained with label smoothing. Can you report these results for Tables 2, 3. I suspect that a noticeable amount of improvement can be obtained by using label smoothing, especially if undistillable classes are semantically similar classes (See [2, 3]).

5) CUB200 KD results are missing (Not available in Supplementary as well).

This is an interesting paper. In my assessment, I feel that the weaknesses of this paper outweigh the strengths. But I’m happy to change my opinion based on the rebuttal.


====================

**Post-Rebuttal:**

Thank you authors for the great effort on the rebuttal.

I have increased my recommendation. Authors have addressed most of my concerns although concrete understanding of undistillable classes still remains unclear.

As discussed, please include details on the effect of label smoothing on the proposed TLLM framework.

==================

[1] Müller, Rafael, Simon Kornblith, and Geoffrey E. Hinton. "When does label smoothing help?." Advances in neural information processing systems 32 (2019).

[2] Shen, Z., Liu, Z., Xu, D., Chen, Z., Cheng, K. T., & Savvides, M. (2021). Is label smoothing truly incompatible with knowledge distillation: An empirical study. In ICLR

[3] Chandrasegaran, K., Tran, N. T., Zhao, Y., & Cheung, N. M. (2022). Revisiting Label Smoothing and Knowledge Distillation Compatibility: What was Missing?. ICML

---

> ### Author Response · Authors · 2022-08-02
> **Author Responses to Reviewer NdAQ (Part 2/2)**
>
> R4C4:
> > If the authors use label smoothing for undistillable classes (Lines 66-68) in the student, it is not reasonable to compare with reported Baselines as label smoothing will further improve accuracy (by alleviating models’ overconfidence [1, 2, 3]). A more reasonable comparison will be to include the accuracies of students trained with label smoothing. Can you report these results for Tables 2, 3. I suspect that a noticeable amount of improvement can be obtained by using label smoothing, especially if undistillable classes are semantically similar classes (See [2, 3]).
>
> We appreciate the reviewer's concern about the impact of label smoothing in our method. We share the view that label smoothing is a practical regularization technique that can alleviate the model's overconfidence. Typically, knowledge distillation [4,5,6] can also be considered as a regularization technique, which has similar functionality to label smoothing. Thus, **it is uncommon to perform knowledge distillation with label smoothing together**. In our work, we adopt label smoothing on those undistillable classes mainly because the KD is halted on these classes. The regularization effect brought by KD is removed on the undistillable classes. Thus, we leverage label smoothing to compensate for such a missing regularization effect.
>
> To directly address the reviewer's concern, we ran experiments to compare the baseline in Table 2 with and without label smoothing, as requested. We present the results as follows:
>
> | Method  | ResNet32x4/ResNet8x4 | ResNet50/MobileNetV2 | ResNet50/ShuffleNetV1 |
> |---------|----------------------|----------------------|-----------------------|
> | Student | 72.50%| 64.60%| 70.50%|
> | KD      | 73.08%| 67.28%| 74.07%|
> | LS      | 73.86%| 65.11%| 70.78%|
> | TLLM    | 75.53%| 69.54%| 76.67%|
>
>
> The LS denotes the label smoothing. We set the hyperparameter to 0.1 for all experiments, including TLLM. We can observe that the improvement of LS is marginal, and TLLM achieves significantly better performance compared to LS. We believe these empirical results can support that **the improvement of TLLM is not attributed to the use of label smoothing**.
>
> We are also glad that R4 cited three excellent papers [1,2,3] that study the compatibility of **label-smoothed teacher** and knowledge distillation. We stress that these works study the phenomenon of **when the teacher model is trained with label smoothing techniques** and how to distill the student model with such a label-smoothed teacher. However, TLLM leverages label-smoothing during distilling the student network. **Thus, these works[1,2,3] are orthogonal to our work**. In addition, we have included [2] in our study (Figure 7 in the initial Appendix), which shows that label-smoothed teachers also suffer from undistillable classes.
>
> R4C5
> > CUB200 KD results are missing (Not available in Supplementary as well).
>
> We thank the reviewer for raising this point. It appears that our description of the experiments caused some confusion. We stress that our initial version only evaluates TLLM on CIFAR-100 and ImageNet1K. As quoted in line 54, the experiments on CUB-200 are only used to verify the existence of undistillable classes. In Appendix, our implementation details on CUB-200 primarily correspond to the experiments of how we obtain the results in Figure 4 (top right) and Figure 10 in the Appendix. However, to expand upon our initial submission, we ran additional experiments on CUB-200 with two teacher-student pairs (ResNet50/ResNet18 and ResNet50/MobileNetV2). We compare with KD and present the results as the following:
>
> | Method  | ResNet50/ResNet18 | ResNet50/MobileNetV2 |
> |---------|-------------------|----------------------|
> | Student | 76.89%| 79.20%|
> | KD      | 79.96%| 81.18%|
> | TLLM    | 81.15%| 82.26%|
>
> We also achieved superior performance compared to the KD baseline. We included these experiments along with implementation details in the new section Appendix F.3.
>
> R4C6
> > What is the final optimization/loss function (preferably in equation form)?
>
> Notably, we leverage the change of per-class teacher curve (distillation loss) on the validation dataset to determine whether the distillation process of the particular classes should be terminated. **Our method did not use any extra loss function**.
>
> R4C7
> > What is the mixture parameter () used for label smoothing in undistillable classes?
>
> We use 0.1 for all experiments.
>
>
> [4] Self-Distillation as Instance-Specific Label Smoothing, Neurips 2020
>
> [5] Revisiting Knowledge Distillation via Label Smoothing Regularization, CVPR 2020
>
> [6] Self-Distillation Amplifies Regularization in Hilbert Space, Neurips 2020
>
> [7] Decoupled Knowledge Distillation, CVPR 2022

---

> ### Author Response · Authors · 2022-08-02
> **Author Responses to Reviewer NdAQ (Part 1/2)**
>
> **Comment:**
> We thank reviewer for the comments and reference reviewer with identifier NdAQ as R4. Comment n of reviewr m is denoted as RmCn.
>
> **[R4C1]**
> > Although the work is technically sound, do you think the explanation / definition of undistillable classes is a bit overkilled? Can the following be a more simpler explanation (I will use ImageNet experiment as example): The teacher network is not perfect. That is the top1 accuracy of ResNet-50 teacher is 76.2% (Table 3). Therefore, the teacher tends to make a noticeable amount of errors. If class-wise analysis is performed on the teacher based on misclassified samples, it corresponds to a subset of classes which can be treated as undistillable classes. Therefore, the proposed TLLM avoids such situations by eliminating classes where the teacher's predictions are largely incorrect and replacing them with ground-truth labels (one-hot vectors). Some analysis I’d like to see are: 1) What is the average accuracy of ResNet-50 teacher (Table 3) for regular and undistillable classes? 2) Can you decompose the improvements obtained () in Tables 2, 3 for regular and undistillable classes?
>
> We understand the reviewer's valid concern about the possible overkilled explanation/definition of undistillable classes. We think the reviewer suggests that the undistillable classes are misclassified samples where the teacher's predictions are largely incorrect (we refer to teacher's hard classes, where in the distillation scenario, they are classes with relatively low accuracy on the teacher model). And the main purpose of TLLM is to eliminate teachers' hard classes during distillation and replace them with ground-truth labels.
>
> We respectfully disagree with the reviewer on this point. We ran additional experiments to test whether the teacher's hard classes are equivalent to undistillable classes. We evaluate three varying teacher-student pairs with KD on CIFAR-100. The per-class test accuracy of the teacher and compared with the change of test accuracy for the student with and without KD (the definition of undistillable classes) are recorded. Our results indicate no correlation between teachers' hard classes and undistillable classes. The details of these experiments are added to the new section Appendix E.2.
>
> To directly address the reviewer's concern, we report the average accuracy of ResNet-50 teacher for regular and undistillable classes as follow:
> ResNet50 regular-classes vs ResNet50 undistillable classes: 76.1% vs 76.2%
> We note that the undistillable classes take over one-third of the overall classes. This result again verifies our previous observation that **teacher's hard classes are not undistillable classes**.
>
> Additionally, when decomposing the improvements obtained in Table 2 (ResNet32x4/ResNet8x4 - KD) for regular and undistillable classes, the TLLM improves the undistillable classes by 1.33% and **3.74%**, respectively. It shows that our method indeed **improves the performance of undistillable classes**.
>
> Overall, we believe that these experiments verify our analysis and demonstrate that the explanation/discussion of undistillable classes cannot be simplified based on teacher's misclassified classes.
>
> **[R4C2]**
> > ImageNet results are insufficient to assess the efficacy of proposed TLLM: What is the reason for using only KR (Chen et. al) as the baseline for ImageNet experiments? Can the authors produce a table similar to Table 2 for ImageNet to show the improvements of the proposed TLLM scheme?
>
> Our previous version only reports the performance of TLLM based on KR because KR is a competitive baseline. We want to demonstrate that our approach can achieve noticeable improvement even on a state-of-the-art KD algorithm. We agree with the reviewer that similar experiments as for CIFAR-100 should be present for ImageNet1K. Thus, we ran additional experiments on ImageNet1K based on KD, AT, and SFTN, respectively.
> | Method               | KD   | TLLM + KD | AT   | TLLM + AT | SFTN | TLLM + SFTN |
> |----------------------|-------|-----------|-------|-----------|------|-------------|
> | ResNet32/ResNet18    | 70.7% | 72.1%| 70.7% | 71.5%| 71.1%| 72.0%|
> | ResNet50/MobileNetV1 | 70.5% | 72.0%| 69.5% | 70.6%| 71.5%| 72.8%|
>
> On both teacher-student pairs, our proposed method can improve over multiple distillation baselines.
>
> **[R4C3]**
> > The Baseline performance of MobileNetV2 in Table 3 (ImageNet) seems to be very low. top1 accuracy should be ~72.154% whereas the reported accuracy is only 68.9%. Please clarify. Link to public pytorch models with top1 accuracies: https://pytorch.org/vision/stable/models.html
>
> We thank R4 for careful reviews and for pointing out this incorrect performance. This is a typo, and we actually conducted the experiments based on **MobileNetV1**. We evaluated MobileNetV1 since it is a conventional experiment presented in many previous works, i.e., DKD[7]. We sincerely apologize for this mistake; we fixed it in the revised version.

---

> ### Author Response · Authors · 2022-08-06
> **Looking forward to your reply**
>
> Dear reviewer NdAQ:
>
> We thank you for the precious review time and valuable comments. We have provided all requested experiments and included an analysis of undistillable classes, which we believe have covered your concerns. We hope to discuss further with you whether or not your concerns have been addressed. Please let us know if you still have any unclear parts of our work.
>
> Best,

---

> > ### Comment · Reviewer_NdAQ · 2022-08-08
> > **Reply**
> >
> > Thank you authors for the great effort on the rebuttal. Authors have addressed most of my concerns although concrete understanding of undistillable classes still remains unclear.
> >
> > **Question**: For results reported in R4C4, does TLLM use LS? Do you have the performance of TLLM when no LS is applied (Only use standard cross entropy loss when learning from ground-truth labels in TLLM)?

---

> > > ### Author Response · Authors · 2022-08-08
> > > **Rolling rebuttal clarifications**
> > >
> > > We are glad that our initial rebuttal clarified most of your concerns. In R4C4, the TLLM use LS. Here, we provide an ablation study which **remove LS from TLLM**. The experiments are similar to table in R4C4. The experiments results are the following:
> > >
> > > | Method  | ResNet32x4/ResNet8x4 | ResNet50/MobileNetV2 | ResNet50/ShuffleNetV1 |
> > > |---------|----------------------|----------------------|-----------------------|
> > > | Student | 72.50%| 64.60%| 70.50%|
> > > | KD      | 73.08%| 67.28%| 74.07%|
> > > | LS      | 73.86%| 65.11%| 70.78%|
> > > | **TLLM w/o LS** | 75.11% | 69.17% | 76.13% |
> > > | **TLLM w/ LS**   | 75.53%| 69.54%| 76.67%|
> > >
> > > The experimental results between TLLM w/o LS" and "TLLM w/ LS " demonstrate that adopting LS in TLLM only improves the overall performance **marginally**. This is because it serves only as a replacement for the regularization effect, which has been removed when undistillable classes are discarded. However, when LS is removed from TLLM (TLLM w/o LS), we can still observe a noticeable improvement in accuracy compared to KD. It again verifies that the **performance gain indeed comes from TLLM** itself.
> > >
> > > Also, we admitted that the concrete understanding of undistillable classes remains unclear. Nevertheless, we believe our work provides an interesting and practical new perspective on understanding the “large teacher worse student” phenomenon and knowledge distillation in general.

---

> > > > ### Comment · Reviewer_NdAQ · 2022-08-09
> > > > **Reply 2**
> > > >
> > > > Thank you for additional results. I have increased my recommendation, although I stand by my review regarding the lack of concrete understanding of undistillable classes.

---

### Official Review · Reviewer_6Hiv · 2022-07-10

**Rating:** 6
**Confidence:** 4
**Soundness:** 3 good
**Presentation:** 3 good
**Contribution:** 3 good

**Summary:**

This paper aims to study the problem of “larger teacher, worse student” in knowledge distillation (KD).
From a data-centric perspective, this paper claims that undistillable classes are the cause of the inefficacy of large teachers. Furthermore, this paper proposes a new KD framework called Teach Less, Learn More (TLLM) to address the issue. In detail, TLLM tries to identify the undistillable classes during training with a moving window and lets the student learn these classes directly from the ground truth labels.

**Questions:**

The explanation from per-class accuracy in general novel. Nevertheless, the above weaknesses refrain me to give a better score. I would be happy to improve my rating as long as the authors address the above weaknesses in the rebuttal.

**Limitations:**

Yes, this paper clearly discusses the limitations and potential negative social impacts in the supplementary.

**Strengths And Weaknesses:**

Strengths

- The data-centric perspective is novel and interesting in the study of “larger teacher, worse student”, which understands the problem much better than the previous capacity mismatch explanation. Moreover, To the best of my knowledge, the per-class accuracy has not been explored in KD before. It deserves to be studied more in the future since the undistillable classes are universal.

- Experiments are comprehensive and the results are convincing. This paper considers many teacher-student pairs and even some advanced architectures such as vision transformers (ViTs). Moreover, this paper considers both output-based KD and feature-based KD.


Weaknesses

- A clearly mathematical definition of the “undistillable class” is missing.
From section 2.1, we know that the “undistillable classes” are classes with bad distillability. However, a specific mathematical distillability measurement ||c is not given, which makes it difficult to understand the detailed implementation in this paper.  As in Figure 5, we know that the undistillable class is a class whose accuracy after KD is lower than that before KD. However, it is unclear whether this is a necessity and sufficiency of undistillable class. It is also unclear about what specific criterion this paper uses to select samples in Figure 1. To this end, this paper should give a clear definition of it in the rebuttal letter.


- More studies of the undistillable classes should be discussed.
This paper mainly studies the proportion of undistillable classes, which is insufficient. More studies about the distribution of these classes should be explored. For example, whether these undistillable classes are challenging classes (i.e., classes with relatively low accuracy before KD), and whether these classes are network-independent.

---

> ### Author Response · Authors · 2022-08-02
> **Author Responses to Reviewer 6Hiv**
>
> **Comment:**
>
> We thank reviewer for the comments and reference reviewer with identifier 6Hiv as R3. Comment n of reviewr m is denoted as RmCn.
>
>
> **[R3C1]**
> > A clearly mathematical definition of the “undistillable class” is missing. From section 2.1, we know that the “undistillable classes” are classes with bad distillability. However, a specific mathematical distillability measurement ||c is not given, which makes it difficult to understand the detailed implementation in this paper. As in Figure 5, we know that the undistillable class is a class whose accuracy after KD is lower than that before KD. However, it is unclear whether this is a necessity and sufficiency of undistillable class. It is also unclear about what specific criterion this paper uses to select samples in Figure 1. To this end, this paper should give a clear definition of it in the rebuttal letter.
>
> We thank the reviewer for raising this point. To help understand the detailed implementation of our work, we provide a formal definition of the distillability for the measurement |c| based on the measurement of accuracy in the revision. We note that this is also the specific criterion that we use to select samples in Figure 1. Indeed, we cannot guarantee that our distillability measurement is a necessary and sufficient condition for the undistillable class. Overall, our work aims to explain the "large teacher worse student" phenomenon based on the empirical observation that there exist some classes in the student model which achieve inferior performance after distillation compared to their vanilla-trained counterpart. We hope our work can inspire the community to explore the necessary and sufficient conditions to define such a phenomenon in a rigorous manner.
>
>
>
>
> \
> **[R3C2]**
> > More studies of the undistillable classes should be discussed. This paper mainly studies the proportion of undistillable classes, which is insufficient. More studies about the distribution of these classes should be explored. For example, whether these undistillable classes are challenging classes (i.e., classes with relatively low accuracy before KD), and whether these classes are network-independent.
>
> We share the sentiment with the reviewer that more studies of the undistillable classes should be discussed. The reviewer also rightly asked whether these undistillable classes are challenging classes for teachers and whether these classes are network-independent. We compare the per-class accuracy of teacher model and student model with the change of accuracy of student model with and without KD. Our experiments cover three teacher-student pairs. The analysis shows that **undistillable classes are not equivalent to challenging classes**. Furthermore, we look into the undistillable class of three varying teacher-student pairs and observe that **undistillable classes are network-dependent**.
>
> To strengthen our work, we conduct an in-depth analysis of undistillable classes. Specifically, we discussed whether different KD methods could result in different undistillable classes on the same teacher-student pair. We further show that undistillable classes are distillation-dependent. We also verify that undistillable classes are hard classes for students. In other words, we cannot determine whether a class is distillable based on either the teacher's prediction or the student's prediction. We hope our study can help readers to understand the observed phenomenon. We included these additional analyses in the new section Appendix E.

---

> > ### Comment · Reviewer_6Hiv · 2022-08-08
> > **Thanks for your reply**
> >
> > Thanks for your reply. The additional analysis of the undistillable classes from the Supplementary is interesting and solves my concerns. I believe this finding can inspire the community to understand KD better. Thus, I keep my positive rating.

---

### Official Review · Reviewer_WWqA · 2022-07-11

**Rating:** 5
**Confidence:** 4
**Soundness:** 3 good
**Presentation:** 3 good
**Contribution:** 3 good

**Summary:**

This paper propose that some classes is not distillable to students, and by removing these classes, the distillation performance would be improved.

**Questions:**

Would the author analyze why some undistillable classes are consistent across multiple experiments, while others are not? And in those consistent classes, is the same group of instances undistillable?

**Limitations:**

Yes

**Strengths And Weaknesses:**

Pros:
  1. The paper is well written and easy to follow.
  2. The idea is simple and effective in practice. The observation that some classes are not distillable to the student is novel.
  3. Experiment result is competitive against some mainstream KD methods.

Cons:
  1. The paper lacks deep insight into these bad distillable classes. The method of this paper is similar to "On the efficacy of knowledge distillation". Both learn teachers at the beginning, and then gradually start to learn one-hot labels. The difference is that this paper argues that certain classes are the cause of the problem. So why do hard examples come in the form of categories, are all samples of these certain classes hard to distill?

---

> ### Author Response · Authors · 2022-08-02
> **Author Responses to Reviewer WWqA**
>
> We thank reviewer for the comments and reference reviewer with identifier WWqAas R2. Comment n of reviewr m is denoted as RmCn.
>
>
> **[R2C1]**
> > The paper lacks deep insight into these bad distillable classes. The method of this paper is similar to "On the efficacy of knowledge distillation". Both learn teachers at the beginning, and then gradually start to learn one-hot labels. The difference is that this paper argues that certain classes are the cause of the problem. So why do hard examples come in the form of categories, are all samples of these certain classes hard to distill?
>
> We think the hard examples come in the form of categories because of the nature of supervised learning. The goal of supervised learning is to maximize inter-class distance. As a result, the feature representation of samples that are belonged to the same classes could be clustered together during training, and thus the hard samples form into categories. Furthermore, we confirm that not all samples are hard-to-distill.
>
> We share the sentiment with the reviewer that there is room for improvement in terms of interpretability in the undistillable classes. Therefore, we perform an in-depth analysis to understand the undistillable classes. Specifically, we show that undistillable classes depend on teacher-student pairs and KD methods. We also verified that undistillable classes are not equivalent to hard classes in either the teacher model or the student model. Finally, we included a new section, Appendix E.
>
> \
> **[R2C2]**
> > Would the author analyze why some undistillable classes are consistent across multiple experiments, while others are not? And in those consistent classes, is the same group of instances undistillable?
>
> We thank the reviewer for raising these points. We realized the word "consistent," as mentioned in the initial submission, is not rigorous. In our revision, we reorganized this part and added new experiments, which analyze the undistillable classes in diverse teacher-student pairs and distillation methods more thoroughly. Specifically, we observe undistillable classes depending on distillation methods and network. We only observe only five consistently undistillable classes under the different capacity gaps between the teacher and student model. Within these classes, we observe a tiny group of instances that are consistently undistillable. We believe these instances share no common property, and these samples are sensitive to knowledge distillation.

---

> > ### Comment · Reviewer_WWqA · 2022-08-08
> > **Reply**
> >
> > The author addresses some of my concerns. This paper is an improvement on ES that uses a subset of the dataset, specifically a subset of classes, to distill students. Nonetheless, there remains a lack of explanation as to why the undistillable subset appear as classes when undistillable classes lack consistency across different models and methods.

---

> > > ### Author Response · Authors · 2022-08-08
> > > **Rolling rebuttal clarification**
> > >
> > > We are glad that our initial rebuttal addressed some of your concerns. Here, we are focusing on the parts (primarily three points) where we recon more justification is necessary to clarify the point that our approach is not an improvement of ESKD.
> > > &nbsp;
> > > 1. The **motivations** behind ESKD and TLLM are fundamentally different.
> > >
> > > ESKD shows that teacher with higher accuracy does not necessarily make better students. They conclude this is a **consequence of mismatched capacity**. Different from ESKD, our work attributes the “large teacher worse student” phenomenon to the **existence of undistillable classes** from a **data-driven standpoint**.
> > >
> > > &nbsp;
> > > 2. Different motivation leads to varying methods.
> > >
> > > Because ESKD thinks reducing the capacity gap between teacher and student is the key to resolving the “large teacher worse student” issue, they propose to leverage the early-stopping strategy to **train teacher model** and then adopt this **early-stopped teacher** to perform distillation on student model. They assume that applying regularization (early-stopping in their case) to teacher can reduce the capacity gap (highlighted in **Section 5.5 from the ESKD paper**.) Instead, our approach is to adopt class-wise early-stopping, which aims to **eliminate the adverse effect** brought by undistillable classes.
> > >
> > > Furthermore, we stress that ESKD proposed two early-stopping strategies. The first is to apply early-stopping during distillation (the methodologically similar one to TLLM), and the second is to apply early-stopping on teacher. Note that the former (the one reviewer think is methodologically similar to TLLM) **does not** change the observation that, quoted in Section 5.3 of ESKD, “larger, more accurate teachers do not result in more accurate students.” In contrast, we demonstrate that TLLM allows **larger teachers to make better students** (Section 4.2) and perform significantly better than ESKD (see R1C1 for details). It highlights the importance of our motivation and the difference in our methods, which makes, at first glance, a methodologically similar approach, much more effective than ESKD.
> > > &nbsp;
> > >
> > > 3. Our contribution is more than TLLM
> > >
> > > We also emphasize that our contribution is more than TLLM. As we have stated in our paper, we examine the “large teacher worse student” phenomenon from a data-centric perspective and provide some interesting observations on undistillable classes that are well supported by extensive experiments. Our proposed technique, despite being simple, is very effective in partially resolving this issue. In our revision, we included analysis to help readers to gain some understanding of the characteristics of these classes. We believe that our work offers a novel perspective on the analysis of knowledge distillation.</li>

---

> ### Author Response · Authors · 2022-08-06
> **Looking forward to your reply**
>
> Dear reviewer WWqA :
>
> We thank you for the precious review time and valuable comments. We have a detailed analysis of the undistillable class in our revision. We hope to discuss further with you whether or not your concerns have been addressed. Please let us know if you still have any unclear parts of our work.
>
> Best,

---

### Official Review · Reviewer_jTkD · 2022-07-12

**Rating:** 5
**Confidence:** 4
**Soundness:** 3 good
**Presentation:** 3 good
**Contribution:** 3 good

**Summary:**

This paper introduced a new data-centric perspective on the phenomenon of 'larger teacher, worse student’.  They search for the classes that disrupt the distillation process, they defined them as the undistillable classes, and then exclude them to improve the performance of the small student network when distillation; this framework is called 'Teach Less Learn More (TLLM)’. Specifically, to find the undistillable classes, they suggest two criteria; per class accuracy and agreement prediction agreement. Various ablation studies support their observation and experimental results on multiple datasets with varying networks validate the effectiveness of TLLM.

**Questions:**

I was wondering if there is a practical way to apply TLLM over the new dataset.

**Limitations:**

Please check the weaknesses and questions; in terms of practicality and analysis.

**Strengths And Weaknesses:**

### Strengths
- The observations are interesting and well supported experimentally.

- The method is conceptually simple and seems to work well across the board.

- Experiments are conducted with several neural network architectures and various datasets.

### Weaknesses
- The concept and approach of the proposed method are different from ESKD[3] but somewhat similar in methodology for improving the KD performance; ESKD also halts distillation from teacher to student at the last phase. A direct comparative experiment with ESKD will enhance the completeness of this paper.

- The proposed method, TLLM, is potentially impactful, but it should use significantly more computational resources and time for finding undistillable classes. Thus, this point is not useful from a practical point of view. It would be great if they could suggest an efficient way to find undistillable classes.

- They claim that unstillable classes are universes, but there is no basis for that. It is necessary to analyze the characteristics of the classes and why they interfere with distillation.

- Minor point
> Line127, replace j with M
> Line163-5, please check the suitability of examples
> Line166, what is the meaning of crystal
> Figure 4, explain the meaning of the dashed line

---

> ### Author Response · Authors · 2022-08-02
> **Author Responses to Reviewer jTkD**
>
> **Comment:**
>
> We thank reviewer for the comments and reference reviewer with identifier jTkD as R3. Comment n of reviewr m is denoted as RmCn.
>
> **[R1C1]**
> > The concept and approach of the proposed method are different from ESKD[3] but somewhat similar in methodology for improving the KD performance; ESKD also halts distillation from teacher to student at the last phase. A direct comparative experiment with ESKD will enhance the completeness of this paper.
>
> We agree with the reviewer's point that direct comparison with ESKD will enhance the completeness of this paper. We ran experiments on both CIFAR-100 and ImageNet1K to present a fair comparison with ESKD as follows:
>
>
> | Method | CIFAR100: ResNet50/MobileNetV2 | ImageNet1K: ResNet32/ResNet18 |
> |--------|--------------------------------|-------------------------------|
> | KD     | 64.6%                          | 72.1%                         |
> | ESKD   | 66.4%                          | 72.0%                         |
> | TLLM   | **69.5%**                          | **72.1%**                         |
>
> The results show a clear advantage of our approach. Notably, ESKD sets a hard early-stopping threshold during distillation and stops the entire distillation processing once the threshold is met. Compared to their naive strategy, which uses a hard threshold, our method tracks the per-class distillation loss and dynamically stops the distillation process in a class-wise manner, which encapsulates ESKD. The empirical study also supports our claim that early stopping on per-class distillability is critical to students' performance. We will include these results in the Appendix.
>
>
> \
> **[R1C2]**
> > The proposed method, TLLM, is potentially impactful, but it should use significantly more computational resources and time for finding undistillable classes. Thus, this point is not useful from a practical point of view. It would be great if they could suggest an efficient way to find undistillable classes.
>
> We appreciate the reviewer’s concern regarding the computational cost at train time and suggest that our approach may not be useful from a practical point of view. We think it is clear that knowledge distillation methods do not bring extra computational costs at test time. We want to stress that this is critical in most real-world scenarios where offline training resources are abundant, yet online computation budgets are tight.
>
> We also agree with the reviewer’s insightful concern that overwhelming extra train-time computational cost may prevent using our method in practice. We point out that the **extra computational cost of TLLM is small**. Specifically, our method adopts the **same total training epochs** as the baseline KD methods. Our approach indeed needs to access the validation distillation loss with two extra forward-pass (**without backpropagation**) for each epoch. Generally, the most computational cost during training comes from backpropagation. Also, the **validation dataset is typically much smaller than the training dataset**. Therefore, TLLM would not introduce overwhelming computational costs compared to the baseline KD approaches.
>
> \
> **[R1C3]**
> > They claim that unstillable classes are universes, but there is no basis for that. It is necessary to analyze the characteristics of the classes and why they interfere with distillation.
>
> To clarify, we conclude that the undistillable classes universally exist based on our extensive experiments on multiple datasets, teacher-student pairs, and distillation methods. We agree with the reviewer that it is necessary to analyze the characteristics of the undistillable classes and understand why they interfere with distillation. Therefore, we perform an in-depth analysis to understand the undistillable classes. Specifically, we show that undistillable classes depend on teacher-student pairs and KD methods. We also verified that undistillable classes are not equivalent to hard classes in either the teacher model or the student model. We included a new section Appendix E.
>
>
> \
> **[R1C3]**
> > Minor point 1) Line127, replace j with M
> 2) Line163-5, please check the suitability of examples
> 3) Line166, what is the meaning of crystal
> 4) Figure 4, explain the meaning of the dashed line
>
> We thank the reviewer for raising these points. We fixed 1) in our revision, we removed the unsuitable example as mentioned in 2), we replaced the word "crystal" with the word "obvious" (the current sentence is "the distilled student model shows obvious class-dependent bias") in 3), and we added an explanation of the dashed line (the average increased accuracy after distillation).

---

> ### Author Response · Authors · 2022-08-06
> **Looking forward to your reply**
>
> Dear reviewer jTkD:
>
> We thank you for the precious review time and valuable comments. We have provided corresponding responses, including a comparison with ESKD and an analysis of undistillable classes, which we believe have covered your concerns. We hope to further discuss with you whether or not your concerns have been addressed. Please let us know if you still have any unclear parts of our work.
>
> Best,

---

### Author Response · Authors · 2022-08-06
**Rebuttal revision**

We would like to thank all the reviewers for their insightful comments and constructive suggestions. Below we provide our detailed response to each reviewer. We took into account all comments and suggestions to prepare the revised version, with major changes highlighted in blue (there are also other minor changes on wording and typos).

To summarize the most important aspects:

- The new section Appendix E directly addresses the concern of R1C3 (of reviewer id: jTkD), R2C1 (of reviewer id: WWqA), R3C2 (of reviewer id: 6Hiv), and R4C1 (of reviewer id: NdAQ). We provide an analysis of the undistillable classes to understand the characteristics of these classes. Specifically, we discuss the correlation between hard classes of both teacher and student models) with undistillable classes. We investigate whether undistillable classes depend on teacher-student pairs, capacity gaps, and distillation methods. We also provide a visualization to help the reader gain a more intuitive understanding of undistillable classes.

- In the new section Appendix F, we included more experiments on ImageNet (requested by R4C2, of reviewer id: NdAQ), direct comparison with label smoothing method (requested by R4C4, of reviewer id: NdAQ), experiments on CUB-200 (requested by R4C5, of reviewer id: NdAQ), and direct comparison with ESKD (requested by R1C1, of reviewer id: jTkD).

- We included a more formal definition of the distillability in Section 2.1 to address the concern raised by R3C1 (of reviewer id: 6Hiv).

We thank the reviewers again for their time and valuable suggestions!

---

### Meta-Review · Area_Chair_Jdtw · 2022-08-26

**Recommendation:** Accept
**Confidence:** Certain

**Metareview:**

This paper makes an interesting observation on knowledge distillation such that excluding certain undistillable classes improves performance. This observation is quite interesting and potentially impactful for a better understanding of knowledge distillation. The authors use this observation to consistently improve the existing knowledge distillation methods in the experiments. One weakness is that the explanation of "why" it is beneficial to exclude certain classes is not very satisfactory.

Nevertheless, the strength of this paper outweighs the weakness. All the reviewers are positive about this paper and I also recommend acceptance.

**Award:**

No

---

### Decision · Program_Chairs · 2022-09-14

Accept